# Theoretical Applicability of Different Occluder Systems for Entry Closure in Type B Aortic Dissection: An Image-Morphological Study

**DOI:** 10.3390/biomedicines13102338

**Published:** 2025-09-24

**Authors:** Miroslav Yordanov, Alexander Oberhuber, Johannes Frederik Schäfers, Raman Anzz, Abdulhakim Ibrahim

**Affiliations:** Department of Vascular and Endovascular Surgery, University Hospital Muenster, Albert-Schweitzer-Campus 1, 48149 Muenster, Germany

**Keywords:** type B aortic dissection (TBAD), dSINE (distal stent graft-induced new entry), occluder system, celiac artery (CA), left subclavian artery (LSA), FET (Frozen Elephant Trunk)

## Abstract

**Objective:** Type B aortic dissection is a life-threatening medical condition. Endovascular closure of the primary entry by means of TEVAR is considered, nowadays, the gold standard if operative treatment is necessary. The aim of this study is to analyse the theoretical applicability of selective endovascular entry sealing using different occluder systems. **Methods:** A CT-graphic analysis of 102 patients who received TEVAR from January 2017 to June 2023 was performed. Patients with an intramural haematoma were excluded. The study patients were divided in two groups: type B aortic dissection (*n* = 87) and distal stent graft-induced new entry (*n* = 15). The TBAD group included patients with acute (*n* = 63), subacute (*n * = 12), and chronic aortic dissections (*n* = 12). The CTA analysis of the location, length, and width of the entry was performed using Aquarius iNtuition (TeraRecon, Inc., Foster City, CA, USA). After completion of the data collection, the possible application of all three occluder systems (ASD-Occluder, Septal-Occluder, and Amplatzer™-Occluder) was analysed, with reference to the Instructions for Use. **Results:** The ASD-Occluder from GORE is produced in five different sizes. It can be used in 81.4% (*n* = 83) of all patients in the overall study, including 82.8% TBAD (*n* = 72) and 73.3% of dSINE (*n* = 11) patients. When using the ASD-Occluder, 10.3% (*n* = 9) of patients are expected to have complete vascular coverage of the LSA based on our CTA analysis. The Septal-Occluder from GORE is offered in three different sizes. Complete entry closure can theoretically be achieved in fifty patients (57.5%) with TBAD and in nine patients (60%) with dSINE, based on CTA analysis and IFU criteria. With the use of the Septal-Occluder, 3.9% (*n* = 4) of the dSINE patients and 4.6% (*n* = 4) of the TBAD patients were expected to have complete aortic branch occlusion. The Amplatzer™-Occluder from Abbott is provided in 27 different sizes to effectively seal defects with a diameter of 4 to 56 mm. It can technically be used in 90.1% of patients (*n* = 92), of which 89.7% with TBAD (*n* = 78) and 93.3% with dSINE (*n* = 14) to completely seal the entry. **Conclusions:** CTA analysis in patients with TBAD and dSINE demonstrated that by the theoretical application of occluder systems, a seal of the entry would be achieved in 57.8% to 90.1% of the patients. However, in addition to entry closure, the use of occluder systems can also lead to unintentional partial (10.7–23.5%) or complete (3.9–22.5%) coverage of adjacent aortic branches. The clinical significance and applicability of the occluder system should be reviewed in future studies and practical applications to evaluate safety, efficacy, and possible complications in order to define the benefit–risk balance.

## 1. Introduction

The aortic dissection arises through an intimal tear, also known as the “primary entry” and can lead to a life-threatening aortic rupture or malperfusion of vital organs and/or extremities. Endovascular therapy is the treatment of choice for complicated TBAD and is achieved by implantation of an aortic stent graft that occludes the primary entry. By the following perfusion reduction in the false lumen (FL), the aortic wall is stabilised, and thrombosis of the false lumens is induced. Conventional surgery is an alternative in the case of contraindications for TEVAR [1].

TEVAR allows a non-selective sealing of the entry; however, the possible early postoperative complications include stroke (6%), 30-day mortality (0.5–7.1%), retrograde aortic dissection (0.5–1.6%), spinal ischemia (0–3.4%), mesenteric ischemia (6%), and vascular access problems [2,3,4]. The late complications include endoleak, post-dissection aneurysms (7.9%), stent graft dislocation, chronic renal insufficiency, and dSINE (distal stent graft-induced new entry) [2,3].

The aim of this study was to assess the theoretical applicability of three different types of endovascular occluder systems to achieve a selective sealing of the entry in patients with type B aortic dissection. We estimated the potential rate of undesirable partial or complete occlusion of adjacent aortic branches, e.g., the left subclavian artery (LSA) or the celiac artery (CA), as well as how many spinal and/or lumbar arteries, otherwise occluded by conventional TEVAR, would still be perfused by application of occluder system and consequently reduce the risk of spinal ischemia.

## 2. Materials and Methods

This is a retrospective, single-centre study. Between January 2017 and June 2023, 102 consecutive patients with acute (*n* = 63), subacute (*n* = 12), and chronic type B aortic dissection (*n* = 12), as well as dSINE (*n* = 15), were included in the study. Patients with intramural haematoma were excluded. The cohort was divided in two groups. In the first group, the primary entry of TBAD was treated in terms of TEVAR. The second group involved patients with dSINE after TEVAR had been performed. The implantation of occluder systems was evaluated separately as a new possible procedure for both groups (TBAD and dSINE). The study was conducted in accordance with the Declaration of Helsinki. The study protocol was approved by the local ethics committee (Ethik-Kommission Muenster, protocol number 2019–764-b-S).

### 2.1. Image Analysis

The analysis is based on the preprocedural contrast-enhanced CT images.

In order to detect the precise location of the entry (Figure 1), representation in a 3D reconstruction is essential. An analysis software (Aquarius iNtuition, Version 4.4, TeraRecon, Inc., Foster City, CA, USA) was used to perform 3D reconstructions with the data from the CT. An important tool in the analysis software is the creation of the centreline (CPR: Curve Planar Reconstruction), which serves as an imaginary axis and enables the measurement of distances in the 3D onstruction.

We analysed the entry length, width, and distance to the adjacent large aortic branches (celiac artery or left subclavian artery). The centreline is an imaginary line or axis that runs throughout the aorta along the blood flow. Typically, the entry is longitudinal or oblique to the aortic axis, and to measure the entry length, the centreline was adjusted from the proximal to the distal entry and measured accordingly (Figure 2A). The centreline was manually adapted so that it runs exactly in the middle of the undissected aorta and then through the centre of entry (from the true lumen to the false lumen, Figure 2B). In this state, the maximal width of the entry was measured exactly in the 3D reconstruction (Figure 2C). The distance to the entry and adjacent vessels (LSA, CA) was also measured from the centreline. In TBAD, the distance between the centre of the entry and LSA was measured, while in dSINE, the distance between the centre of the entry and CA was measured. Both distances are important for analysing the undesirable potential aortic branch coverage when using the occluder system.

Once the data collection was completed, all three occluder systems (ASD-Occluder, Septal-Occluder, and Amplatzer™-Occluder) were reviewed individually for each entry to ensure that the entry would be fully sealed. Using the IFU of the occluder systems, the detailed information (e.g., recommended treatable defect size and maximum outer diameter) was taken into consideration.

Subsequently, individual possible entry occlusions were re-examined to determine whether they would cause a partial or complete coverage of the LSA or the CA. In TBAD patients, LSA coverage is possible, whereas in dSINE patients, due to the anatomical proximity of the corresponding vessel, the coverage of CA is possible.

### 2.2. Devices

Occluders are permanent artificial vascular implants that can occlude vessels, shunt connections, etc., and are inserted endovascularly. They are manufactured from titanium and/or nickel and are available in different sizes. In our study, we considered three types of occluders. The ASD-Occluder [5] (GORE^®^, Flagstaff, AZ, USA) consists of an insertion system and an implantable occluder and is available in five different configurations, with diameters ranging from 8 mm to 35 mm. The Septal-Occluder [6] (GORE^®^, Flagstaff, AZ, USA) differs from the ASD-Occluder in terms of diameter and available sizes. This occluder varies in diameter from 5 mm to 17 mm and is only available in three different sizes. The Amplatzer™-Occluder [7] (Abbott Medical Plymouth, MN, USA) system is made of a nitinol braid and polyester material with diameter ranging from 4 mm to 56 mm. Compared to the previous two occluder systems, it offers an extended range of options and is advantageous in everyday clinical practice (Table 1). See Instructions for Use for complete device information, including approved indications and safety information.

### 2.3. Statistical Analysis

The statistical software SPSS for Windows (version 26.0; IBM Corp., Armonk, NY, USA) was used. For parametric data, continuous variables are represented as means ± standard deviations (SD) and for non-parametric data, as medians with interquartile ranges (IQR). In categorical data, continuous variables are represented as raw numbers and percentages. Student’s *t*-test for normally distributed variables was used for comparisons of continuous variables. For categorical variables, a Friedman test (comparison of three groups) was used. A *p*-value of <0.05 was considered statistically significant.

## 3. Results

Patient characteristics and preoperative data between 01/2017 and 06/2023 in 102 patients were included in this study of which 84 (82.4%) were men. The mean age of the patients was 63.8 ± 13.5 years. The cohort was divided into two groups based on the diagnosis: TBAD or dSINE. The first group included 87 patients, 72 of whom were men (82.7%). The median age was 63.5 ± 13.7 years. The second group included 15 patients with dSINE, 12 of whom were men (80%). The median age was 65.5 ± 12.1 years.

Arterial hypertension was recorded in 93% of patients (*n* = 95), including 93.1% of patients with TBAD (*n* = 81) and 93.3% of patients with dSINE (*n* = 14). Stroke as a pre-existing condition was found in 13.7% of the patients (*n* = 14), including 14.9% with TBAD (*n* = 13) and 6.6% with dSINE (*n* = 1). No significant difference was detected between both groups (Table 2).

In the preoperative CTA, we clearly identified the primary entry or dSINE in all patients who were enclosed and analysed in our study. Based on CTA findings, we determined the entry length, width, and maximum diameter in patients with TBAD and dSINE (Table 3). The results demonstrated that the entry length of dSINE was significantly smaller at 8.9 ± 5 mm than in TBAD 15.4 ± 13.2 mm (*p* = 0.02). The entry width was slightly larger for TBAD (15 ± 11.1 mm) than for dSINE (9.9 ± 7 mm), but the difference was not statistically significant (*p* = 0.071). The maximum diameter also did not differ significantly between both groups (20.03 ± 13.7 mm for TBAD vs. 12.5 ± 7.8 mm for dSINE, *p* = 0.197).

### 3.1. ASD-Occluder from GORE^®^

This device (Figure 3) would be applicable in 82.8% of patients with TBAD (*n* = 72) and in 73.3% of patients with dSINE (*n* = 11) in various sizes to completely close the entry. Based on our measurements, it would lead to a partial coverage of CA in all patients and of LSA in only 5 patients (5.7%). Complete coverage of the LSA would be expected in 9 patients (10.3%), whereas no complete coverage of the CA was estimated (Table 4).

### 3.2. Septal-Occluder from GORE^®^

Based on our CTA analysis, the entry could be closed with the Septal-Occluder (Figure 4) in 59 patients (57.8%). Partial occlusion of LSA in 2.3% of patients with TBAD (*n* = 2) and of CA in 60% of patients with dSINE (*n* = 9) was estimated. On the other hand, complete coverage of the LSA would be expected in 3.9% of the overall cohort (*n* = 4), with all patients coming from the TBAD group. No complete coverage of the CA was to be expected in any groups. There was no significant difference between both groups (Table 5).

### 3.3. Amplatzer™-Occluder from Abbott

This device (Figure 5) is offered in 27 different sizes to effectively seal defects with a diameter from 4 to 56 mm. According to our analysis, the Amplatzer™ could technically be applied in 90.1% of patients (*n* = 92), including 89.7% of patients with TBAD (*n* = 78) and 93.3% of patients with dSINE (*n* = 14) in order to achieve complete closure of the entry. Partial aortic branch coverage would occur in 24 patients (23.5%) (12 patients with TBAD (13.8%) and 12 patients with dSINE (80%)). A complete coverage of the LSA would be expected in 22.5% (*n* = 23), in contrast to CA, where no complete coverage is to be expected, according to the measurements (Table 6).

Subsequently, the applicability of the three available occluder systems was examined separately. Table 7 and Table 8 demonstrate the technical applicability of the different occluder systems for treatment of TBAD and dSINE, providing a comprehensive overview and a more detailed presentation of the study results.

Based on the CTA analysis, the ASD-Occluder could be used most often in the primary entry tear group without LSA coverage (*n* = 58/87, 66.6%, *p* < 0.001), while the Amplatzer™-Occluder could be used most often, in 78 patients (89.6%) if LSA coverage was to be neglected (Table 8. In the dSINE group, no significant difference between both groups could be detected.

## 4. Discussion

In patients with TBAD, the endovascular closure of the entry by means of TEVAR is considered the gold standard. However, for secure entry occlusion, a sufficient proximal landing zone is required. If such zone is not available, a branched TEVAR procedure in the aortic arch could still be performed, though mainly with custom-made devices, which cannot be delivered quickly. On the other hand, a major open surgery such as a Frozen Elephant Trunk (FET) operation is not appropriate for elderly and polymorbide patients. In such cases, entry occlusion by means of endovascular implantation of occluder systems could be a very attractive alternative. In this study we analysed the theoretical applicability of three different occluder systems (ASD-Occluder, Septal-Occluder, and Amplatzer™-Occluder), considering their IFU only in terms of defect size. Our study demonstrated that in 92 patients (90.1%), the Amplatzer™-Occluder can be applied to close the primary entry in TBAD. The ASD-Occluder ranks second and can technically be applied in 83 patients (81.4%) and the Septal-Occluder in turn can be used in 59 patients (57.8%). One argument for the preference of Amplatzer™-Occluder lies in the extensive range of the available sizes. In addition to effectively close the entry, our goal is to ensure the further perfusion of adjacent aortic branches, such as the LSA and the CA, which could be partially or completely covered by the occluder. CTA analysis shows that Septal-Occluder can lead to partial vascular coverage (LSA or CA) in 11 patients (10.7%) and complete vascular coverage in 4 patients (3.9%). The Amplatzer™-Occluder can cause partial vascular coverage in 24 patients (23.5%) and complete vascular coverage in 23 patients (22.5%).

According to the IFU, the ASD-Occluder and the Septal-Occluder from GORE are permanently implanted devices indicated for the percutaneous, transcatheter closure of ostium secundum atrial septal defects (ASD). In our study we estimate the theoretical applicability for entry closure in TBAD, considering the entry size as a single indicator for the device applications. However, according to the IFU, there must be adequate room in the atrial chambers to allow the right and left atrial discs to lie flat against the septum. In our study we did not consider the room in the true and false lumen to ensure the adequate application of the device. The Amplatzer™ Septal Occluder from Abbott is a percutaneous, transcatheter atrial septal defect closure device. The IFU clearly states the importance of the preprocedural invasive defect sizing using a round compliant balloon to choose the appropriate device. In patients with a fragile dissection membrane (acute and subacute TBAD), this could lead to undesirable entry enlargement with potential hazardous effects on the aortic hemodynamic.

When applying occluder systems, it is very important to take into consideration the acuity of the aortic dissection. As we know, the dissection membrane in acute and subacute thoracic aortic dissection (TBAD) is very fragile, in contrast to chronic dissection. Theoretically, the endovascular application of occluders in patients with an unstable dissection membrane (i.e., acute or subacute TBAD) could lead to enlargement or the creation of new entries, which could eventually worsen the aortic haemodynamic. In contrast, the hypertrophied dissection membrane (chronic TBAD) provides a stable base for occluder system application. To our knowledge, studies reporting endovascular entry occlusion in terms of the endovascular application of occluder systems have only concerned patients with chronic TBAD.

The use of smaller introducers, e.g., for ASD-Occluder (10–14 Fr) or for a Septal-Occluder (12 Fr), offers an advantage compared to a TEVAR, which requires large-calibre introducers (16–24 Fr). Therefore, it is fair to assume that the rate of access complications would be lower. In endovascular treatment of the aorta, general anaesthesia is usually preferred for precise deployment. In the use of occluder systems for the treatment of congenital heart defects such as atrial septal defect (ASD) in cardiology, only local anaesthesia is usually sufficient. Whether this would be applicable to the selective closure of an aortic entry remains an open question [5,6,7,8,9,10].

Due to the stiffness of the prosthesis, TEVAR leads to an increased afterload of the left ventricle as well as an increase in systolic pressure amplitudes of up to 17%. Potential complications may include a postoperatively increase in aortic diameter, the development of a secondary aneurysm in the descending aorta, and the need for reintervention. On the other hand, the use of occluder systems can help maintain the elasticity of the aorta and thus avoid the Windkessel effect [11,12,13,14,15,16]. Nevertheless, the risk of entry enlargement or the creation of new entries during occluder application in patients with a fragile dissection membrane (acute or subacute TBAD) outweighs the benefits of this endovascular treatment. There are currently no studies available to confirm our hypothesis, while occluder systems have only been used in isolated cases. If the occluder systems do not achieve the desired effect or if the therapy fails, the endovascular operation in terms of TEVAR can still be performed to achieve the desired treatment goal [11,12,13,14,15,16,17].

In one study, the occluder systems and the established TEVAR therapy were compared in six patients with chronic aortic dissection. In three patients, the dissection was treated by thoracic stent graft. In another three patients, the entry was sealed with an Amplatzer™-Occluder, ASD-Occluder, or Septal-Occluder. The treatment with occluders resulted in a shorter hospital stay of 4 ± 1 days compared to conventional therapy of 11 ± 6 days. In the follow-up after five years, no further reinterventions were necessary for these three patients, and corresponding aortic remodelling was observed [18]. Applying the occluders, it is necessary to approach the entry from the false into the true lumen. Thus, the distal shield unfolds in the true lumen and the proximal shield in the false lumen. The release takes place under X-ray with a final angiography to confirm the entry seal.

In another clinical trial from Sweden, 14 patients with chronic AD (6 patients with type A and 8 patients with type B aortic dissection) were treated with occluder systems between 2007 and 2016. In ten patients (*n* = 10/14, 71%), successful sealing of the entries was documented in the CT images over a median follow-up period of 7.6 years [19]. Four of these ten patients required reintervention later to achieve complete closure. For the remaining four patients, complete sealing of the entries was not achievable, which is why early intervention in terms of FET or TEVAR was performed.

In contrast to other studies concerning entry occlusion by means of the endovascular application of occluder systems, in our study, we analyse the rate of the unintentional theoretical obstruction of adjacent aortic branches, e.g., left subclavian artery or celiac artery, which could lead to potentially lethal complications [19].

## 5. Limitations

The main limitation of this study is its theoretical design. Our results are based only on computer models and are therefore not directly comparable with the results of well-established operative approaches applied by patients with TBAD. Due to the lack of prospective validation, our results could only be compared with single-centre data, presenting only a small number of patients.

## 6. Conclusions

Clinical data is only available from two small studies, which demonstrated mediocre results. The clinical significance and applicability of the occluder system have significant potential and should be reviewed in future studies to evaluate safety, efficacy, and possible complications to clarify the benefit–risk balance.

## Figures and Tables

**Figure 1 biomedicines-13-02338-f001:**
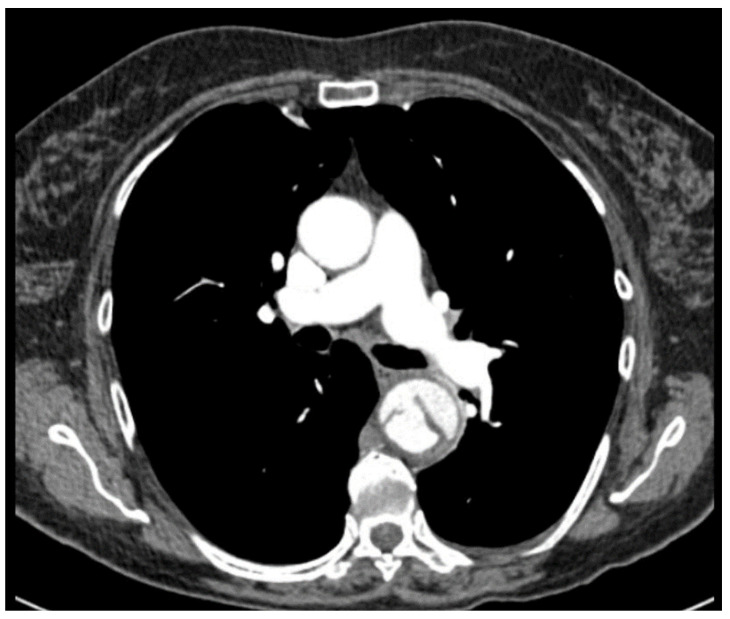
CT chest of a type B aortic dissection with the primary entry.

**Figure 2 biomedicines-13-02338-f002:**
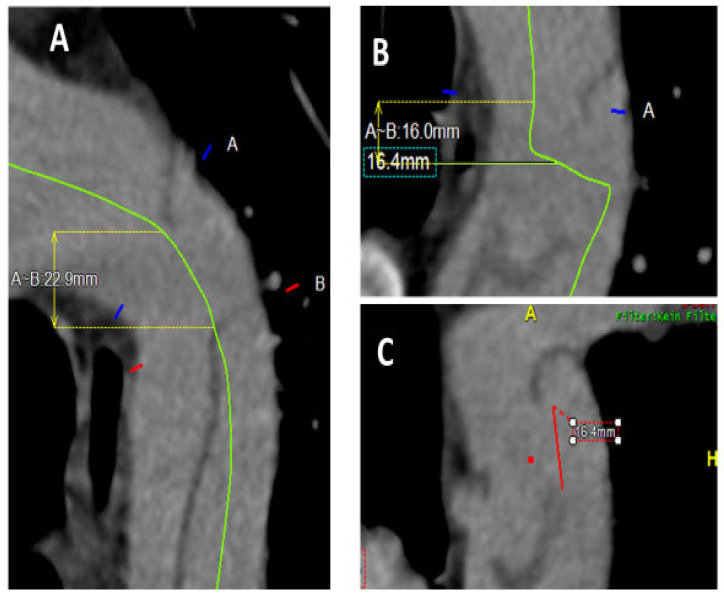
Descending thoracic aorta after transfer to analysis software (Aquarius iNtuition, Version 4.4, TeraRecon, Inc., Foster City, CA, USA. (**A**) Green centreline to measure entry length from proximal to distal entry. (**B**) Vertical course of the centreline through the dissection membrane (entry centre) (green line). (**C**) Red line: measurement of the maximal entry width in 3D reconstruction.

**Figure 3 biomedicines-13-02338-f003:**
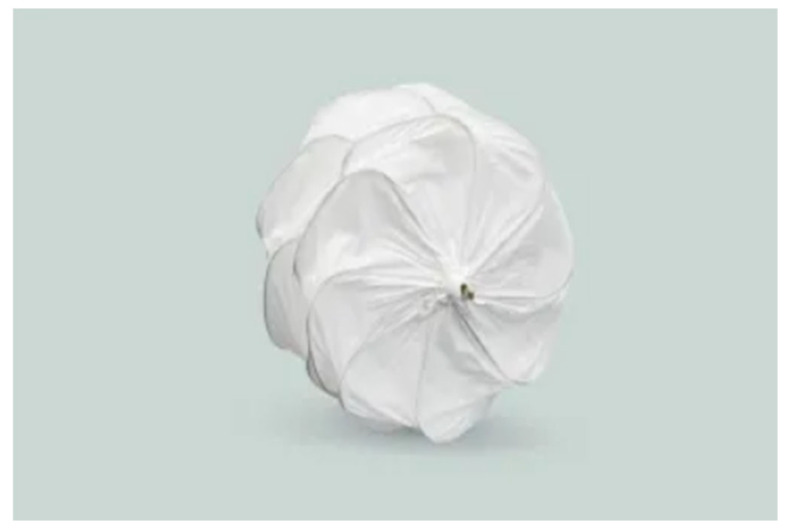
GORE^®^ CARDIOFORM^®^ ASD-Occluder.

**Figure 4 biomedicines-13-02338-f004:**
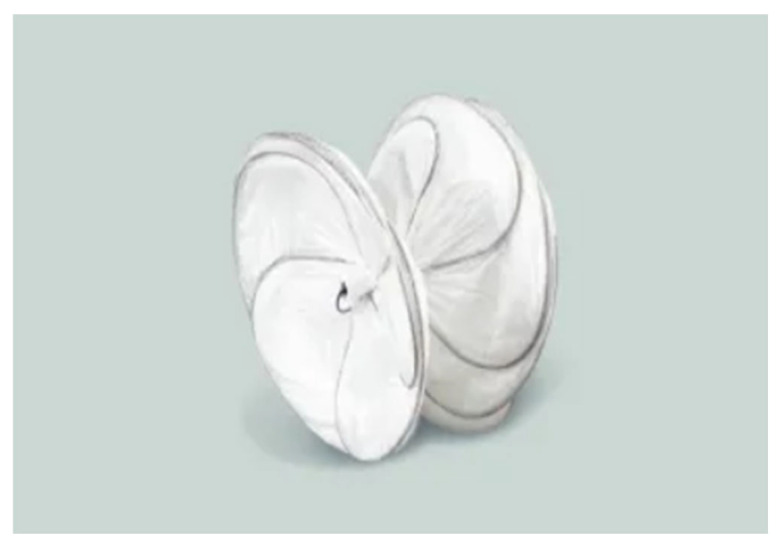
GORE^®^ CARDIOFORM^®^ Septal-Occluder.

**Figure 5 biomedicines-13-02338-f005:**
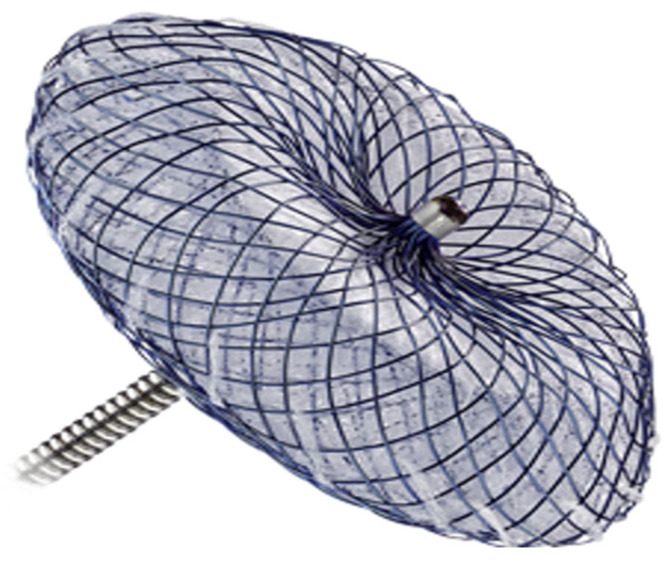
Amplatzer™-Occluder from Abbott. Courtesy of Abbott. © 2025. All rights reserved.

**Table 1 biomedicines-13-02338-t001:** Comparison of the three occluder systems.

	ASD-Occluder from GORE^®^	Septal-Occluder from GORE^®^	Amplatzer™ from Abbott
Recommended treatable defect size (mm)	8–35	5–17	4–56
Available sizes (mm)	1–5	1–3	1–27
Material	Platinum-filled nitinol wire framework, PTFE	Platinum-filled nitinol wire framework, PTFE	Nitinol braid and PTFE
Introducer Sheath (Fr)	10–14	10	6–13

**Table 2 biomedicines-13-02338-t002:** Preoperative patient characteristics.

	Total (*n* = 102)	TBAD (*n* = 87)	dSINE (*n* = 15)	*p*-Value
**Demographic aspects**				
Age ± standard deviation	63.8 ± 13.5	63.5 ± 13.7	65.5 ± 12.1	0.511
Men (*n*, %)	84 (82.4%)	72 (82.7%)	12 (80%)	0.531
**Classification in terms of acuity (*n*, %)**				
Acute	63 (61.8%)	63 (72.4%)		0.411
Subacute	12 (11.8%)	12 (13.8%)		0.553
Chronic	27 (26.4%)	12 (13.8%)	15 (100%)	0.421
**Pre-existing conditions, *n* (%)**				
Arterial hypertension	95 (93.1%)	81 (93.1%)	14 (93.3%)	0.605
Diabetes mellitus	12 (11.7%)	81 (93.1%)	2 (13.3%)	0.481
Coronary artery disease	16 (15.6%)	15 (17.2%)	1 (6.6%)	0.437
Stroke	14 (13.7%)	13 (14.9%)	1 (6.6%)	0.701
Marfan syndrome	6 (5.8%)	6 (6.8%)	0	0.371
Nicotine consumption	13 (12.7%)	11 (12.6%)	2 (13.3%)	0.511
Chronic kidney disease	15 (14.7%)	14 (16%)	3 (20%)	0.478

**Table 4 biomedicines-13-02338-t004:** ASD-Occluder from GORE^®^.

	Total (*n* = 102)	TBAD (*n* = 87)	dSINE (*n* = 15)	*p*-Value
**Applicability (*n*, %)**				
Yes	83 (81.4%)	72 (82.8%)	11 (73.3%)	0.293
No	19 (18.6%)	15 (17.2%)	4 (26.7%)	
**Various ASD-Occluder sizes 8–35 mm, (*n*, %)**				
Nr. 1 (8–15 mm)	49 (48%)	40 (46%)	9 (60%)	
Nr. 2 (13–20 mm)	65 (63.7%)	54 (62%)	11 (73.3%)	
Nr. 3 (18–25 mm)	72 (70.5%)	61 (70.1%)		
Nr. 4 (23–30 mm)	78 (76.4%)	67 (77%)		
Nr. 5 (28–35 mm)	83 (81.4%)	72 (82.8%)		
**Vascular coverage (*n*, %)**		**LSA**	**CA**	
Partial	16 (15.7%)	5 (5.7%)	11 (73.3%)	0.288
Complete	9 (8.8%)	9 (10.3%)	0	

**Table 5 biomedicines-13-02338-t005:** Septal-Occluder from GORE^®^.

	Total (*n* = 102)	TBAD (*n* = 87)	dSINE (*n* = 15)	*p*-Value
**Applicability (*n*, %)**				
Yes	59 (57.8%)	50 (57.5%)	9 (60%)	0.544
No	43 (42.1%)	37 (42.5%)	6 (40%)	
**Various Septal-Occluder sizes (5–17 mm) (*n*, %)**				
Nr. 1 (5–11 mm)	23 (22.5%)	20 (23%)	3 (20%)	
Nr. 2 (11–14 mm)	38 (37.2%)	33 (38%)	5 (33.3%)	
Nr. 3 (14–17 mm)	59 (57.8%)	50 (57.5%)	9 (60%)	
**Vascular coverage (*n*, %)**		**LSA**	**CA**	
Partial	11 (10.7%)	2 (2.3%)	9 (60%)	0.506
Complete	4 (3.9%)	4 (4.6%)		

**Table 6 biomedicines-13-02338-t006:** Amplatzer™-Occluder from Abbott.

	Total (*n* = 102)	TBAD (*n* = 87)	dSINE (*n* = 15)	*p*-Value
**Applicability (*n*, %)**				
Yes	92 (90.1%)	78 (89.7%)	14 (93.3%)	0.549
No	10 (9.8%)	9 (10.3%)	1 (6.7%)	
**Vascular coverage (*n*, %)**		**LSA**	**CA**	
Partial	24 (23.5%)	12 (13.8%)	12 (80%)	0.201
Complete	23 (22.5%)	23 (26.4%)	0	

**Table 7 biomedicines-13-02338-t007:** Applicability of different occluder systems (type B aortic dissection).

Type B Aortic Dissection (*n* = 87)				
	ASD-Occluder from GORE^®^	Septal-Occluder from GORE^®^	Amplatzer™ from Abbott	*p*-Value
Use without LSA/CA coverage	58/87 (66.6%)	44/87 (50.5%)	43/87 (49.4%)	<0.001
Use with possible partial LSA/CA coverage	5/87 (5.7%)	2/87 (2.3%)	12/87 (13.8%)	0.002
Use with possible LSA/CA complete coverage	9/87 (10.3%)	4/87 (4.6%)	23/87 (26.4%)	0.007
Technically not possible	15/87 (17.2%)	37/87 (42.5%)	9/87 (10.3%)	0.001

**Table 3 biomedicines-13-02338-t003:** Entry size for TBAD and dSINE.

	TBAD	dSINE	*p*-Value
Entry length in mm (mean value ± standard deviation)	15.4 ± 13.2	8.9 ± 5	0.02
Entry width in mm (mean value ± standard deviation)	15 ± 11.1	9.9 ± 7	0.071
Maximum diameter in mm (mean value ± standard deviation)	20.03 ± 13.7	12.5 ± 7.8	0.197

**Table 8 biomedicines-13-02338-t008:** Applicability of different occluder systems (dSINE).

dSINE (*n* = 15)				
	ASD-Occluder from GORE^®^	Septal-Occluder from GORE^®^	Amplatzer™ from Abbott	*p*-Value
Use without LSA/CA coverage	0/15 (0%)	0/15 (0%)	2/15 (13.3%)	0.40
Use with possible partial LSA/CA coverage	11/15 (73.3%)	9/15 (60%)	12/15 (80%)	0.368
Technically not possible	4/15 (26.7%)	6/15 (40%)	1/15 (6.7%)	0.042

## Data Availability

The original contributions presented in this study are included in the article. Further inquiries can be directed to the corresponding author.

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
