# Peer review of "Theoretical Applicability of Different Occluder Systems for Entry Closure in Type B Aortic Dissection: An Image-Morphological Study"

_biomedicines, 2025, doi:10.3390/biomedicines13102338_

Round 1

Reviewer 1 Report

Comments and Suggestions for Authors

Thank you for submitting this manuscript to Biomedicines. I found this manuscript, which predicts the feasibility of closing the entry tear in type B aortic dissection using ASD closure devices based on CT image analysis, to be interesting. The estimation of closure feasibility, such as by calculating sealable area, was particularly engaging.

That said, I have several concerns regarding the clinical applicability. First, in ASD closure, the device is anchored on the relatively firm atrial septum, whereas in aortic dissection, especially in the acute and subacute phases, the intimal flap is extremely fragile and unstable. Previous clinical reports have all been in the chronic phase, when the intimal flap has thickened and become more durable, making closure less problematic. In contrast, in the acute or subacute phases, even if device landing were technically possible, it is doubtful whether sufficient anchoring width or tissue strength could be achieved. Deployment of a device that generates resistance to aortic blood flow could create new tears around the device in this setting.

This is precisely why TEVAR is suitable in the acute and subacute phases: it provides circumferential support to stabilize fragile intimal flaps and, being tubular, it offers less flow resistance and less device instability. If sealing the entry tear alone were sufficient, this would indeed be a promising strategy. However, clinical outcomes are generally better when tears are treated in the subacute or early chronic phase, before intimal thickening occurs. Waiting until the chronic phase, when the intimal flap has stabilized, risks progressive enlargement of the aortic hiatus and uncontrolled dilation of false lumen of the thoracic descending aorta, which would be counterproductive.

While I acknowledge the advantages the authors mention, such as a lower risk of spinal cord ischemia and applicability in patients with very small access vessels, I believe that, in the acute or subacute phase, the risks of entry closure by these devices outweigh the benefits. I encourage the authors to address these concerns more explicitly in the Discussion.

In summary, this is an innovative and thought-provoking concept that may provide valuable insights for future device development. However, its immediate clinical applicability appears limited, and further refinement and validation will be required before this approach can be safely translated into practice.

Author Response

Comment 1:  That said, I have several concerns regarding the clinical applicability. First, in ASD closure, the device is anchored on the relatively firm atrial septum, whereas in aortic dissection, especially in the acute and subacute phases, the intimal flap is extremely fragile and unstable. Previous clinical reports have all been in the chronic phase, when the intimal flap has thickened and become more durable, making closure less problematic. In contrast, in the acute or subacute phases, even if device landing were technically possible, it is doubtful whether sufficient anchoring width or tissue strength could be achieved. Deployment of a device that generates resistance to aortic blood flow could create new tears around the device in this setting.

Answer 1: Thank you very much for your comments. I totally agree that there is a difference between the acute and chronic aortic dissection in terms of membrane stability/ fragility. Taking your comment into consideration I have added a new paragraph (page 9-10, paragraph 3, line 261-270) in the Discussion section:

`` When applying occluder systems, it is very important to take into consideration the acuity of the aortic dissection. As we know, the dissection membrane in acute and subacute thoracic aortic dissection (TBAD) is very fragile, in contrast to chronic dissection. In theory, the endovascular application of occluders in patients with unstable dissection membrane (i.e. acute or subacute TBAD) could lead to enlargement or the creation of new entries, which could eventually worsen the aortic haemodynamic. In contrast, the hypertrophied dissection membrane (chronic TBAD) provides a stable base for occluder system application. To our knowledge, studies reporting endovascular entry occlusion in terms of the endovascular application of occluder systems have only concerned patients with chronic TBAD.``

Comment 2: This is precisely why TEVAR is suitable in the acute and subacute phases: it provides circumferential support to stabilize fragile intimal flaps and, being tubular, it offers less flow resistance and less device instability. If sealing the entry tear alone were sufficient, this would indeed be a promising strategy. However, clinical outcomes are generally better when tears are treated in the subacute or early chronic phase, before intimal thickening occurs. Waiting until the chronic phase, when the intimal flap has stabilized, risks progressive enlargement of the aortic hiatus and uncontrolled dilation of false lumen of the thoracic descending aorta, which would be counterproductive.

Answer 2: Thank you very much for your comment. I agree with your oppinion. In our aortic centre we perform a large ammount of TEVAR, many of which in acute setting. 

Comment 3: While I acknowledge the advantages the authors mention, such as a lower risk of spinal cord ischemia and applicability in patients with very small access vessels, I believe that, in the acute or subacute phase, the risks of entry closure by these devices outweigh the benefits. I encourage the authors to address these concerns more explicitly in the Discussion.

Answer 3: Thank you very much for pointinh it out. Taking this into account, I have added a new sentence in the last paragraph of the Discussion (page 10, paragraph 5, line 284-287):

`` Nevertheless the risk of entry enlargement or creation of new entries during occluder application in patients with fragile dissection membrane (acute or subacute TBAD) outweighs the benefits of this endovascular treatment. ``

Comment 4: In summary, this is an innovative and thought-provoking concept that may provide valuable insights for future device development. However, its immediate clinical applicability appears limited, and further refinement and validation will be required before this approach can be safely translated into practice.

Answer 4: Thank you very much for the appropriate, professional und very interesting comments.

Reviewer 2 Report

Comments and Suggestions for Authors

I carefully read the article titled " Theoretical Applicability of Different Occluder Systems for Entry Closure in Type B Aortic Dissection: An Image-Morphological Study" sent to me for review. My comments, criticisms, and suggestions are listed below:

  1. TEVAR is an important treatment strategy for type B aortic dissections complicated by malperfusion, rupture, or rapid dilatation of the distal arch or proximal descending aorta. In patients with unoccluded distal re-entry, incomplete false lumen thrombosis and pseudoaneurysmal dilatation due to unrestricted retrograde flow can lead to adverse outcomes. In such cases, the use of occluders has emerged as a viable alternative to ensure complete closure of the false lumen and coverage of any associated re-entry.

  1. In this study, the authors aimed to analyze the theoretical feasibility of selective endovascular access sealing using different occlusive systems. I would like to congratulate the authors for this well-planned and executed work.
  2. If more than one re-entry is detected or if a re-entry is missed during the initial procedure, I recommend commenting on the techniques to be applied and the types of occluders to be used.
  3. It would be good to have a discussion with literature on eliminating complications that may occur during the use of occluders.
  4. Similarly, the discussion on adjunctive reinterventions will also increase the contribution of the article to the literature.
  5. What are the authors' comments regarding the smallest and largest entry diameters that occluders can be used in? It would be appropriate to comment on occluder selection and any technical modifications in these situations.
  6. What are the differences in approaches to acute, subacute and chronic dissections and what are the possible complications and solutions?
  7. What are your thoughts on alternative anesthesia methods to general anesthesia? It would be helpful if you could comment on this in the discussion section.

Once again, I congratulate the authors for this excellent work.

Best Regards

Author Response

Dear reviewer,

Thank you very much for your professional comments.

Comment 1: TEVAR is an important treatment strategy for type B aortic dissections complicated by malperfusion, rupture, or rapid dilatation of the distal arch or proximal descending aorta. In patients with unoccluded distal re-entry, incomplete false lumen thrombosis and pseudoaneurysmal dilatation due to unrestricted retrograde flow can lead to adverse outcomes. In such cases, the use of occluders has emerged as a viable alternative to ensure complete closure of the false lumen and coverage of any associated re-entry.

Answer 1: Thank you very much. I totally agree with this statement.

Comment 2: In this study, the authors aimed to analyze the theoretical feasibility of selective endovascular access sealing using different occlusive systems. I would like to congratulate the authors for this well-planned and executed work.

Answer 2: Thank you very much, it is very kind of you. 

Comment 3: If more than one re-entry is detected or if a re-entry is missed during the initial procedure, I recommend commenting on the techniques to be applied and the types of occluders to be used.

Answer 3: Thank you for your comment. Regarding the re-entry our tactic is to wait and regularly perform a CT-Angiography to estimate the remodelling of the aortic wall. If a progress of the postdissection aneurysm is detected, we would induce a false lumen thrombosis by endovascular insertion of a pillow or would perform a TEVAR to occlude the re-entry.

Comment 4: It would be good to have a discussion with literature on eliminating complications that may occur during the use of occluders.

Answer 4: Thank you, this is a very good point. We had thought about that but there are only very limited publications regarding the use of occluders and there is very scarce information in terms of the possible complications. We have not performed this operation in our clinic and accordingly have no experience whit it. That is why we prefered not to make any discussions on possible complications. 

Comment 5: Similarly, the discussion on adjunctive reinterventions will also increase the contribution of the article to the literature.

Answer 5: Thank you very much. I find it reasonable, however our study is based on theoretical suggestions, and we have not performed the procedure on real patients. As we have no clinical experience with the procedure, we did not want to comment on reinterventions.

Comment 6: What are the authors' comments regarding the smallest and largest entry diameters that occluders can be used in? It would be appropriate to comment on occluder selection and any technical modifications in these situations.

Answer 6: Thank you very much for pointing this out. In Table 1. we compared the recommended treatable defect size (mm) for the three occluder systems according to the official IFU. As we wanted to stay completely within the IFU for each of the three occluder systems, we did not want to make any suggestions for technical modifications.

Comment 7: What are the differences in approaches to acute, subacute and chronic dissections and what are the possible complications and solutions?

Answer 7: Thank you for this absolutely reasonable question. We address it in a new paragraph (page 9-10, paragraph 3, line: 261-270) in the discussion:

`` When applying occluder systems, it is very important to take into consideration the acuity of the aortic dissection. As we know, the dissection membrane in acute and subacute thoracic aortic dissection (TBAD) is very fragile, in contrast to chronic dissection. In theory, the endovascular application of occluders in patients with unstable dissection membrane (i.e. acute or subacute TBAD) could lead to enlargement or the creation of new entries, which could eventually worsen the aortic haemodynamic. In contrast, the hypertrophied dissection membrane (chronic TBAD) provides a stable base for occluder system application. To our knowledge, studies reporting endovascular entry occlusion in terms of the endovascular application of occluder systems have only concerned patients with chronic TBAD.``

Comment 8: What are your thoughts on alternative anesthesia methods to general anesthesia? It would be helpful if you could comment on this in the discussion section.

Answer 8: 

  1. Thank you for the comment. We have only mentioned this topic in the discussion of the original article (page 10, paragraph 4, line: 274-278):

`` In endovascular treatment of the aorta, general anaesthesia is usually preferred for precise deployment. In the use of occluder systems for the treatment of congenital heart defects such as atrial septal defect (ASD) in cardiology, only local anaesthesia is usually sufficient. Whether this would be applicable to the selective closure of an aortic entry, remains open. ``

It would definitely be of benefit for the patient if the operation could be performed in local anaesthesia, but we do not have a real experience with this operation and therefore did not want to make further comments.

Thank you once again for the appropriate, professional und very interesting comments.

Reviewer 3 Report

Comments and Suggestions for Authors

this paper discusses the theoretical use of plugs to occlude the entry tear in type B dissection. I have the following questions/comments:

  • the IFU for each device should be discussed. the paper seems to imply that using plugs in this setting would be within the device IFU.
  • - please comment on anticipated aortic remodeling with this approach. 
  • consider including pictures of the described devices
Comments on the Quality of English Language

Needs improvement for clarity

Author Response

Dear reviewer, 

Thank you very much for your comments.

Comment 1: The IFU for each device should be discussed. the paper seems to imply that using plugs in this setting would be within the device IFU.

Answer 1: Thank you very much for your appropriate comment.

I have changed the second sentence in the first paragraph of the Discussion (page 9, paragraph 1, line 234-236) as follows: `` In this study we analysed the theoretical applicability of three different occluder systems (ASD-Occluder from GORE, Septal-Occluder from GORE and Amplatzer™ Septal Occluder from Abbott), considering their IFU only in terms of defect size.  ``

Additionally, I have included a new paragraph in the Discussion (page 9, paragraph 2, line 248-260):

´´According to the IFU, the ASD-Occluder and the Septal-Occluder from GORE are permanently implanted devices indicated for the percutaneous, transcatheter closure of ostium secundum atrial septal defects (ASD). In our study we estimate the theoretical applicability for entry closure in TBAD, considering the entry size as a single indicator for the device applications. However, according to the IFU there must be adequate room in the atrial chambers to allow the right and left atrial discs to lie flat against the septum. In our study we did not consider the room in the true and false lumen to ensure the adequate application of the device. The Amplatzer™ Septal Occluder from Abbott is a percutaneous, transcatheter atrial septal defect closure device. The IFU clearly states the importance of the preprocedural invasive defect sizing using a round compliant balloon in order to choose the appropriate device. In patients with fragile dissection membrane (acute and subacute TBAD) this could lead to undesirable entry enlargement with potential hazardous effects on the aortic hemodynamic. ´´

Comment 2: Please comment on anticipated aortic remodeling with this approach. 

Answer 2: Thank you for the interesting question. The goal of this treatment strategy is to induce a false limen thrombosis through entry occlusion and thereafter a remodelling of the aortic wall. Our study is based on a theoretical model, and we have not performed the operation on real patients, due to which we have not any observations regarding the aortic wall remodelling. It is to assume, that the remodelling will be induced through entry occlusion, however this should be proved in further studies.

Comment 3: Consider including pictures of the described devices.

Answer 3: Thank you very much for the good idea. Accordingly, I have added pictures of the devices in the article.

Thank you very much for the appropriate, professional und very interesting comments.